# The limited screening accuracy of the Patient Health Questionnaire-2 in detecting depression among perinatal women in Italy

**Antonella Gigantesco**[1]*, **Gabriella Palumbo**[1], **Loredana Cena**[2], **Laura Camoni**[1], **Alice Trainini**[2], **Alberto Stefana**[2], **Fiorino Mirabella**[1]

**1** Center for Behavioural Sciences and Mental Health, National Institute of Health, Rome, Italy, **2** Department of Clinical and Experimental Sciences, Section of Neuroscience, Observatory of Perinatal Clinical Psychology, University of Brescia, Brescia, Italy

* antonella.gigantesco@iss.it

**Data Availability Statement:** All relevant data are within the paper and its Supporting Information files.

## Abstract

### Background

The PHQ-2 was recently recommended by the International Consortium for Health Outcomes Measurement as a form of initial perinatal screening, followed by the EPDS only for women with positive PHQ-2 score. However, the accuracy of the PHQ-2 in perinatal clinical practice has been barely researched, to date. In the present study, we aim to assess the accuracy of the PHQ-2 against the EPDS in a large sample of perinatal women.

### Methods

A total of 1155 consecutive women attending eleven primary or secondary health care centres throughout Italy completed the EPDS and the PHQ-2 during pregnancy (27-40-weeks) or post-partum (1-13-weeks). Sensitivity, specificity, positive predictive value (PPV), negative predictive value (NPV), positive likelihood ratio, negative likelihood ratio, post-test probabilities and area under the curve (AUC) of the PHQ-2, using a cut-off of $\geq 3$, were calculated.

### Main findings

During pregnancy, PHQ-2 revealed low sensitivity (39.5%) and PPV (39.4%) but high specificity and NPV (97.5%). In postpartum, it revealed very low sensitivity (32.7%) and moderately high NPV (80.9%), but high specificity (99.3%) and PPV (94.4%). Given the low sensitivity despite the high specificity, the PHQ-2 demonstrated poor accuracy (AUC from 0.66 to 0.68).

### Conclusion

Initial screening by means of PHQ-2 failed to identify an acceptable number of perinatal women at-risk of depression in Italian clinical practice. The PHQ-2 performance suggested that it has insufficient sensitivity and discriminatory power, and may be inadequate as a screening tool for maternal depression.

**Funding:** The author(s) received no specific funding for this work.

**Competing interests:** The authors have declared that no competing interests exist.

## Introduction

In Italy, the proportion of women who experience depression during pregnancy and postpartum ranges from 6% [1] to 12% [2, 3] and from 7 to 20% [1, 4], respectively. Perinatal depression has been associated with poorer outcomes such as a reduction in a woman's ability to perform daily activities and parenting [5], which increases the risk of significant adverse consequences over the years for the psychological well-being and health of the family [6]. It should be highlighted that maternal depression has a negative impact on the child's development [7].

Given that effective psychosocial interventions (e.g., the WHO Thinking Healthy Programme) [8] and psychological therapies (e.g., cognitive behavioural therapy and interpersonal therapy) are available [9], the importance of early detection of perinatal depression cannot be overstated. Primary and non-psychiatric specialty care visits represent important opportunities to detect or launch treatment of perinatal depression because, typically, perinatal depressed women are more likely to seek care in obstetrics/gynaecology medical settings than in specialised mental health settings. In light of the points mentioned above, several authors have called for identification procedures for perinatal depression to be introduced in clinical practice such as in obstetrics/gynaecology, paediatrics, or internal/family medicine settings [10, 11].

This demands guidelines regarding the implementation of a standardised approach to detect depression with satisfactory accuracy. It has been observed that the accuracy of depression recognition in non-psychiatric health care settings is unsatisfactory [12] and, to date, there is considerable variability regarding how the ascertainment of the risk of depression should be conducted among perinatal women in non-psychiatric clinical practice [13].

International recommendations have adopted various positions concerning the implementation of depression screening in clinical practice. In spite of the fact that the Australian Clinical Practice Guidelines for perinatal depression [14] declare that the use of a universal screening approach is good practice, and also other countries such as the United States [15] and Canada [16] recommend universal screening using the Edinburgh Postnatal Depression Scale (EPDS) [17], neither the Scottish [18], nor British [19] guidelines recommend its use. However, they do recommend selective sequential testing through a 2-step approach in which a 2-question ultra-brief instrument (i.e., Whooley questions) [20] is administered first, followed, only for some women, by the Patient Health Questionnaire-9 (PHQ-9) [21] or EPDS [17]. It should be noted, however, that the NICE recommendation to use the Whooley questions [19] was made in the absence of any validation studies in a large perinatal population. In fact, there is limited evidence for the use of the Whooley questions as a screening tool for maternal depression, only having been validated in small perinatal populations with a wide range of sensitivities found, between 46 and 100% [22].

Consistently with the recommendation of using a universal screening approach, recently, the International Consortium for Health Outcomes Measurement (ICHOM) identified the 2-item PHQ-2 as an initial step to recognize *at-risk* women to be administered at first contact during pregnancy and again during the early postpartum period, followed by the longer EPDS as confirmatory screening only for women who respond positively to either question [23].

In order to achieve a maximum likelihood that all women at risk of depressive symptoms would be administered the EPDS, it is essential that the PHQ-2 is highly sensitive and, more generally, highly accurate. However, to date, perinatal evidence of PHQ-2 accuracy is limited, and only few studies have validated the PHQ-2 in perinatal settings [22, 24]. Of these, only two have compared the PHQ-2 to a structured diagnostic interview, one showing low sensitivity and moderate specificity [25] and the other moderately high sensitivity and specificity [26]. Apart from these studies, some other studies have been conducted using the EPDS as a

reference standard, with mixed results, since sensitivity ranged from 19% to 100% and specificity ranged from 75% to 93% [27–30].

Given the paucity and the inconsistency of reports, additional research in larger and different maternity populations has been recommended in order to validate the PHQ-2 as part of a maternal health care policy to detect perinatal depression [22, 24, 29].

In the present study, we aimed to test the accuracy of the PHQ-2, using the EPDS as a reference standard, in a large sample of pregnant and postpartum women attending several primary or obstetric-gynaecology secondary care centres throughout Italy. To our knowledge, no study has yet compared the use of the PHQ-2 to the EPDS in Italian perinatal clinical practice. The main goal was to determine whether or not the PHQ-2 was highly sensitive and specific, and therefore able to rule out and rule in depression in maternity clinical practice.

## Methods

### Outline of the study

The study is part of a larger body of work conducted by the Observatory of Perinatal Clinical Psychology (University of Brescia, Italy) and the Italian National Institute of Health. This work included a study which merges a cross-sectional study and a pre–post intervention cohort study [31] with two main objectives: (1) to evaluate the prevalence of both maternal antepartum and postpartum depression and anxiety in a sample of women in Italy (cross-sectional study component) and (2) to evaluate the effectiveness of psychological intervention [32] for both antenatal and postnatal depression (pre-post intervention cohort study component). The present study came from the data collected in the cross-sectional study.

### Study protocol

Participants were recruited from eleven publicly-funded primary or obstetric-gynaecology or paediatrics secondary care centres of the Observatory of Perinatal Clinical Psychology (University of Brescia) throughout Italy (Bergamo, Brescia, Mantua and Milan, Lombardy Region; Bologna, Emilia Romagna Region; Florence, Tuscany Region, Novara and Collegno, Piedmont Region; Roma, Lazio Region; Enna, Sicily Region). Evaluation tools were administered once during the pre- or postpartum period, depending on the characteristics of each healthcare centre. The study was approved by the ethics committee of the Healthcare Centre of Bologna (registration number 0077805, dated 6/27/2017) [29].

### Procedure and participants

Participants were recruited during a routine perinatal health check-up or paediatric vaccination appointment at one of the eleven publicly-funded healthcare facilities between 2017–2018. Consecutive women were invited to participate in the study. Specifically, the participation to the study was offered by the obstetricians or gynaecologists or paediatricians of those facilities. All the women approached were provided with a pamphlet developed as part of the study, in which the purpose, aims and methodology of the study were explained. Women who wished to participate provided their personal information (name or phone number to be contacted later) in order to meet up with trained psychologists. These psychologists had attended a training on screening, assessment, and treatment for maternal perinatal mental health problems, developed by the National Institute of Health [4]. Women who definitively agreed to participate signed an informed consent form. Then, they were underwent a semi-structured interview (not a diagnostic interview) led by the trained psychologists to elicit information on current and past maternal experience with psychiatric conditions and use of psychotropic

drugs. Psychiatric conditions included symptoms of anxiety, depression, psychotic symptoms (i.e., delusions and/or hallucinations), non-suicidal self-harm tendencies, suicidal ideation or substance abuse. The inclusion criteria to be enrolled were being able to speak and read Italian well and being a woman aged ≥18 years with a biological baby aged ≤52 weeks. The exclusion criteria were having issues with drug or substance misuse and/or having on-going psychotic symptoms. The enrolled women were then administered the pertaining scheduled self-report assessment tools for data collection (see Tools). All the women completed the interviews and self-report instruments at the facilities; the majority of them on the same day in which they were invited to join. Few women provided their phone numbers to be contacted for arranging a subsequent appointment at the facilities in order to complete the instruments. The psychologists who administered the self-report tools, made sure that they returned fully completed.

## Tools

**Psychosocial assessment form.**   Socio-demographic data were collected at baseline by means of the Psychosocial Assessment Form [4] which addresses socio-demographic characteristics and other information. Socio-demographic characteristics include: age (years), marital status (married or cohabitating; single, separated, divorced or widowed), educational level (primary or illiterate; secondary high school; University), working status (student, homemaker, or unemployed; temporary employee; permanent employee), economic status (several problems; a few problems without specific difficulties; average to high status) and children living at the time of the current pregnancy/birth (yes-no).

**Edinburgh Postnatal Depression Scale (EPDS).**   The EPDS [17, 33] is the most widely used screening tool for depressive symptoms during pregnancy and postpartum [34, 35]. It is a self-rating measure containing 10 items concerning the symptoms of depression such as anhedonia, feelings of guilt, lethargy, sleep disturbance and suicidal ideation occurring in the past 7 days. Each symptom is scored on a four-point Likert. The total score ranges from 0 to 30, with higher scores indicating more severe depressive symptoms.

**Patient Health Questionnaire (PHQ-2).**   The PHQ-2 [36] consists of the first two items of the PHQ-9 (investigating depressed mood and anhedonia). The questions ask: "Over the last 2 weeks, how often have you been bothered by little interest or pleasure in doing things?" and "Over the last 2 weeks, how often have you been bothered by feeling down, depressed, or hopeless?". For each item, the response options are "not at all" (0), "several days" (1), "more than half the days" (2), and "nearly every day" (3). The total score ranges from 0 to 6, with higher scores indicating greater depressive symptoms.

## Statistical analysis

All analyses were conducted using the Statistical Package for Social Science (SPSS) created for Windows, version 26.0. The EPDS was considered the reference standard. The PHQ-2 was analysed using total score at cut-off point 3 or more, as is usually recommended to define the result as positive [37, 38]. The EPDS total score was transformed into a binary variable to indicate positive screening for depression using the cut-off score of ≥ 13 during pregnancy, and ≥ 10 in the postpartum, as recommended by literature [39]. Socio-demographic characteristics of women with positive screening for depression were summarised using descriptive statistics. The chi-square test (or Fisher exact test) were used to test for differences between women with positive screening for depression and women without for each socio-demographic characteristic. Proportion of depression risk according to the EPDS, and positive PHQ-2 responses are presented as frequencies and percentages with 95% confidence intervals.

**Internal consistency.** The internal consistency of the EPDS and the PHQ-2 were assessed. Due to the fact PHQ-2 includes only two items, the mean inter-item correlation (MIC) was adopted.

**Accuracy (or criterion validity).** The screening accuracy of the PHQ-2, defined as sensitivity, specificity, positive predictive value (PPV), negative predictive value (NPV), positive likelihood ratio (LR+), negative likelihood ratio (LR-), and area under the curve (AUC) against the EPDS cut-points using ROC (Receiver Operating Characteristic) analysis was assessed [40]. Positive and negative post-test probabilities were also calculated. The standard error for the area was set as non-parametric with a 95% confidence interval. Area under the curve (AUC) was interpreted as follows: AUC = 0.60–0.70 = poor, 0.70–0.80 = fair, 0.80–0.90 = good, 0.90–1.0 = excellent.

## Results

A total of 1155 women, 71% of those who were asked to participate in the study, filled out both the EPDS and the PHQ-2.

Both groups of pregnant and postpartum women were primarily in their thirties (Table 1). Overall, the majority of them were married or lived with their partner and were well educated. Furthermore, the majority were employed in paid work and only few had serious economic difficulties.

Significant differences between women with positive EPDS and women without were found regarding educational level, economic problems and marital status. Specifically, among pregnant women, positive EPDS was associated with university education level and, among postnatal women, positive EPDS was associated with having several economic problems and not being married or cohabitating with a partner (Table 1).

### Internal consistency

The internal consistency of the EPDS was: α = 0.80 during pregnancy and α = 0.87 following delivery. The PHQ-2 showed acceptable mean inter-item correlations (MIC) both during pregnancy (MIC = 0.38) and postpartum (MIC = 0.45).

### Proportion of minor depression based on EPDS and PHQ-2 positive screen tests

The percentage of antenatal and postnatal women who screened positive for depression as determined by EPDS totalled 4.0% and 25.9%, respectively (Table 2). Positive cases determined through the PHQ-2 was 4.0% during pregnancy and 8.9% postpartum. Having a positive EPDS was found to be higher in the postpartum than during pregnancy. Regarding PHQ-2, the number of cases was also higher in the first trimester following birth.

### Antepartum and postpartum minor depression: PHQ-2 accuracy

The performance of the PHQ-2 with EPDS as a standard criterion during pregnancy and postpartum is presented in Table 3.

During pregnancy, out of 38 women that scored positive on the EPDS, 15 tested positive and 23 negative using the PHQ-2. Among 38 women who screened positive using the PHQ-2, 15 were found to have probable minor antepartum depression as determined by the EPDS.

Of 916 women that scored negative with the EPDS, 893 tested negative with the PHQ-2 and among 916 women who screened negative with the PHQ-2, 893 were found not to have probable minor antepartum depression.

**Table 1. Socio-demographic, pregnancy and delivery characteristics of participants.** Number (valid percentage). P = statistical significance for the comparison between negative EPDS screening (score<13 during pregnancy and < 10 in postpartum) and positive EPDS screening results (score ≥ 13 during pregnancy and ≥ 10 in postpartum).

| | Antenatal | | Postnatal | P |
|---|---|---|---|---|
| | (from 27 to 40-weeks) | | (from 1 to 13-weeks) | |
| | n (%) | EPDS ≥ 13 | n (%) | EPDS ≥ 10 |
| | | n (%) | | n (%) |
| **Total** | 954 | 38 (4.0%) | 201 | 52 (25.9%) |
| | (%) | | (17.4%) | |
| **Age** | | | | |
| 18–29 | 209 (21.9) | 8 (21.1) | 30 (14.9) | 6 (11.5) |
| 30–35 | 453 (47.5) | 19 (50.0) | 79 (39.3) | 16 (30.8) |
| > 35 | 292 (30.6) | 11 (28.9) | 92 (45.8) | 30 (57.7) |
| **Marital status** | | | | |
| Married or cohabitating | 878 (92.6) | 35 (92.1) | 183 (91.5) | 41 (78.8) |
| Single, separated, divorced or widowed | 70 (7.4) | 3 (7.9) | 17 (8.5) | 11 (21.2) §§ |
| **Educational level** | | | | |
| Primary or illiterate | 100 (10.5) | 9 (23.7)§ | 26 (13.0) | 4 (7.7) |
| Secondary | 341 (36.0) | 11 (28.9) | 83 (41.5) | 21 (40.4) |
| University | 507 (53.5) | 18 (47.4) | 91 (45.5) | 27 (51.9) |
| **Economic status*** | | | | |
| Several problems | 58 (6.2) | 5 (13.2) | 16 (8.1) | 9 (17.3)§ |
| A few problems without specific difficulties | 433 (45.9) | 18 (47.4) | 99 (50.0) | 24 (46.2) |
| Average to high status | 452 (47.9) | 15 (39.4) | 83 (41.9) | 19 (36.5) |
| **Working status** | | | | |
| Student, homemaker, or unemployed | 151 (16.0) | 7 (18.4) | 39 (19.6) | 8 (15.7) |
| Temporary employee | 89 (9.4) | 0 (0.0) | 20 (10.1) | 6 (11.8) |
| Permanent employee | 702 (74.6) | 31 (81.6) | 140 (70.4) | 37 (72.5) |
| **Children living at the time of this pregnancy/birth** | | | | |
| No | 798 (83.6) | 28 (73.7) | 137 (68.2) | 40 (76.9) |
| Yes | 157 (16.4) | 10 (26.3) | 64 (31.8) | 12 (23.1) |

*Regarding economic status: Several problems = having debts, difficulty or inability to pay daily expenses and rent; A few problems without specific difficulties = relatively modest standard of living but without particular difficulties; Average high status = home owned, possibility of taking holidays or travelling for pleasure.

§ p<0.05.

§§ p<0.01.

Following delivery, out of 52 women that scored positive on the EPDS, 17 tested positive and 35 negative on the PHQ-2. Among 18 women who screened positive with the PHQ-2, 17 were found to have probable minor postpartum depression.

Out of 149 women that scored negative with the EPDS, 148 tested negative with the PHQ-2 and among 183 women who screened negative with the PHQ-2, 148 were found not to have probable minor postpartum depression.

During pregnancy, the LR+ ranged from moderate (8.9) to high (27.6) and the LR- ranged from weak (0.5) to very weak (0.8) [41].

Following delivery, the LR+ ranged from moderate (6.6) to very high (357.6), with a wide and imprecise confidence interval, and the LR- was weak (from 0.56 to 0.82) [41].

The post-test probability that a woman who tested negative with the PHQ-2 had minor depression was 2.5% during pregnancy and 19.2% postpartum. The post-test probability that a

**Table 2. Proportion of women with probable minor depression ascertained with EPDS, and case identification using PHQ-2.**

| | Pregnancy (from 27 to 40-weeks) | Postpartum (from 1 to 13-weeks) |
|---|---|---|
| | N = 954 | N = 201 |
| | n | n |
| | % (95% CI) | % (95% CI) |
| Screened positive for depression* | 38 | 52 |
| | 4.0 (2.7–5.1) | 25.9 (19.9–32.1) |
| PHQ-2˚ | 38 | 18 |
| | 4.0 (2.8–5.2) | 8.9 (5.0–12.8) |

* Screened positive for depression: EPDS cut-off score ≥ 13 during pregnancy and ≥ 10 postpartum.

˚ PHQ-2 total score cut-off point ≥ 3.

woman who tested positive with the PHQ-2 had minor depression was 39.4% during pregnancy and 94.4% postpartum.

At the recommended cut-off of ≥ 3, screening accuracy of the PHQ-2 was poor (AUC = 0.66–0.68) (Table 4).

## Discussion

To the best of our knowledge, this is the first study which compares the accuracy of the PHQ-2 to the EPDS in a large sample of perinatal women in Italy in order to consider its possible application as a first step screening in perinatal clinical practice.

For the PHQ-2, we used the optimal cut-off point (≥ 3) taking into consideration combined sensitivity and specificity (Youden index), as recommended by recent literature concerning the accuracy of the PHQ-2 in clinical practice [42].

Both the PHQ-2 and the EPDS demonstrated acceptable internal consistency.

In the present study, one of the main findings was that even though the benefit of including a *triage test* in perinatal clinical practice consists of narrowing the number of women who need more extensive evaluation, the PHQ-2 demonstrated that it excessively narrows the percentage of women that would need a subsequent longer assessment to a very low level of 4% of the original group of prenatal women and 8.9% of the original group of postnatal women.

In fact, the PHQ-2 demonstrated that it was poorly sensitive for identifying expectant women and new mothers at risk for depression (39.5% and 32.7%, respectively) indicating that

**Table 3. Performance of the PHQ-2 against the EPDS.**

| | Antenatal | Postnatal |
|---|---|---|
| | (from 27 to 40-weeks) | (from 1 to 13-weeks) |
| Sensitivity (95% CI) | 39.5 (24.0–56.6) | 32.7 (20.3–47.1) |
| Specificity (95% CI) | 97.5 (96.3–98.4) | 99.3 (96.3–99.9) |
| PPV (95% CI) | 39.4 (27.0–53.4) | 94.4 (69.9–99.2) |
| NPV (95% CI) | 97.5 (96.8–98.0) | 80.9 (77.8–83.6) |
| LR+ (95% CI) | 15.7 (8.9–27.6) | 48.7 (6.6–357.0) |
| LR- (95% CI) | 0.62 (0.48–0.80) | 0.68 (0.56–0.82) |
| Positive post-test probability | 39.4 (27.0–53.4) | 94.4 (69.9–99.2) |
| Negative post-test probability | 2.5 (2.0–3.2) | 19.2 (16.4–22.2) |

CI = Confidence Interval.

**Table 4. ROC analysis of the PHQ-2 (cut-off point ≥ 3) with EPDS as criterion standard.**

| Test variable | AUC | SE | Significance (p) | Lower bound | Upper bound |
|---|---|---|---|---|---|
| *Screened positive for depression* | | | | | |
| *(EPDS as criterion standard)* | | | | | |
| Pregnancy (from 27 to 40-weeks) | 0.68 | 0.05 | 0.000 | 0.58 | 0.79 |
| Postpartum (from 1 to 13 weeks) | 0.66 | 0.049 | 0.001 | 0.56 | 0.76 |

many cases of probable depression remained undetected (60.5% and 67.3% false-negative). This sensitivity was substantially lower than that reported in the original PHQ-2 validation study (39.5% and 32.7% compared to 83%) [37] but not very distant from that reported in a recent meta-analysis which assessed the PHQ-2 pooled performance against a gold-standard diagnostic interview [24]. This meta-analysis showed that the sensitivity of the PHQ-2 in identifying major depression in primary care was lower than that reported in the original study at ≥ 3 cut-off point (pooled sensitivity 64%; with the lower boundary of the 95% CI = 46%). Consistently, a more recent systematic review has confirmed that in settings such as primary care or some inpatient and outpatients specialty care, the PHQ-2 was up to 62% sensitive for cut-off score of 3 or greater, in studies that used fully structured interviews as reference standards [43].

In contrast, we found the PHQ-2 to be highly specific (97.5%-99.3%), suggesting a low risk of false positives and response burden. As a consequence of low sensitivity, despite the high specificity, the PHQ-2 ultimately demonstrated poor accuracy, based on the ROC analyses, suggesting that it did not possess any substantial discriminatory ability both during pregnancy and postpartum (AUCs = 0.68 and 0.66, both corresponding to poor accuracy).

Looking at the PPVs, our findings seemed also to indicate a poor performance of the PHQ-2 in pregnancy, given the very low probability that a woman with a positive result indeed had depression (39.4%). In contrast, the combination of a low prevalence, low sensitivity and high specificity resulted in high NPVs (97.5% in pregnancy and 80.9% postpartum), suggesting that the probability that a woman with a negative PHQ-2 result indeed did not have depression was high, especially in pregnancy.

However, these predictive values remained uninformative, at least in prenatal screening, because they are affected by a very low prior prevalence (from 2.7% to 5.1%), which involved the base rate fallacy [44]. This makes the PHQ-2 worthless under the condition of low prevalence, which means that if you intend to use the PHQ-2, you should only apply it in pregnant populations with high prevalence of depression, such as psychiatric populations or other populations at risk of mental disorders. As evidence of this, both PPV (94.4%) and NPV (80.9%) proved to be more informative and acceptable among our sample of postnatal women, who had a higher prevalence of depression.

As pointed out by several authors [41, 45], when sensitivity is low despite high specificity, the use of the *SpPIn* mnemonics *(high **Sp**ecificity, **P**ositive, rules **In**)* is ineffective. In the present study, the *SpPIn* rule is not applicable also because of low prevalence, especially in our antenatal sample. In the same way, the use of the *SnNOut* mnemonics *(high **S**ensitivity, **N**egative, rules **Out**)* is not applicable because of low prevalence and low sensitivity. As a consequence, LRs and post-test probabilities may be probably the best way to evaluate the strength of the PHQ-2 in our clinical centres, where women will generally have depression at an earlier and milder stage (i.e., have a low or moderately high prevalence of depression).

The LR- of 0.62 in pregnancy and the LR- of 0.68 postpartum were weakly indicative of an absence of depression, suggesting limited accuracy in ruling out the risk of depression following a negative result [41]. After testing negative for the PHQ-2, the post-test probability of

having antenatal depression slightly decreased from 4% (pre-test probability) to 2.5%, with a confidence interval ranging from 2.0 to 3.2%. This means that a negative PHQ-2 was still compatible with a 3.2% post-test probability of depression, which clearly was low in absolute terms but unacceptably high in this situation in which the sensitivity was so low. Moreover, despite the weakly indicative LR-, the post-test probability was low because it started from a very low pre-test probability. As evidence of this, among postpartum women, with a higher pre-test probability and similar LR- of 0.68, the post-test probability resulted in a higher point estimate of 19.2% (CI: 16.4–22.2), which may be considered even more unacceptably high in this situation in which the sensitivity was even lower (32.7%). In this situation, the clinician may only feel moderately confident that depression could be ruled out. To be sure, he/she needs to advise women to proceed with other investigations.

The LR+ in the prenatal sample was 15.7 (CI: 8.9–27.6), suggesting that the PHQ-2 ranged from moderately (estimated shift in probability of at least 30%, considering the lower boundary of the CI) to strongly (estimated shift in probability of at least 45%, considering the upper boundary of the CI) indicative of the presence of depression. The post-test probability of having prenatal depression increased from 4.0% (pre-test probability) to 39.4% (CI: 27.0–53.4). Despite this considerable increase, the post-test probability revealed the presence of depression among only about one in 2–3 women with a positive PHQ-2, under the low pre-test probability and sensitivity. This may make the clinician decide to rule in prenatal depression and proceed with a further screening assessment, but with moderate confidence.

The LR+ in the postnatal sample was 48.7 with a very wide confidence interval (95% CI: 6.6–357.0). The probability of having postnatal depression increased from 25.9% to 94.4% (95% CI: 69.9–99.2) revealing that about one in 1–1.4 women with a positive test had depression. Despite the fact the LR+ suffered from a very wide confidence interval, with the lower 95% confidence interval including 6.6, which was moderately indicative of the presence of depression, this may make the clinician decide to rule in women for the presence of postnatal depression and proceed with the second step of a two-stage screening strategy, but with moderate confidence.

The few available comparable studies are mostly consistent with our findings. In particular, our findings are very consistent with those of a recent Australian study which used the same EPDS cut-offs to test the accuracy of the PHQ-2 for probable minor depression [29]. In that study, during pregnancy (36-weeks), sensitivity, specificity, PPV, NPV, LR+ and LR- were 0.50, 0.95, 0.30, 0.98, 9.4 and 0.53, respectively, and at 6-weeks postpartum were 0.39, 0.97, 0.68, 0.91, 14.59 and 0.62, respectively. In the same study, using ROC analysis, the PHQ-2 cut-off $\geq 3$ was found to be poor to fair for minor depression (AUC = 0.68 in pregnancy; AUC = 0.72 postpartum), similarly to our AUC findings of 0.66–0.68. Our findings were also similar to those of a study of Smith and colleagues [25], which reported low PHQ-2 sensitivity (62%). Very consistently with the present study, another postnatal study [27] showed a sensitivity of 43.5% and a specificity of 97.2%, using the same PHQ-2 cut-off point of 3 or more to identify a positive screen with the EPDS (cut-off point of 10 or more).

A limited number of other studies have reported higher sensitivity. Gjerdingen et al., reported that the PHQ-2 had 75% sensitivity, 88% specificity, 24% PPV and 99% NPV, in a sample of postpartum women (0–1 month postpartum) [26]. However, Gjerdingen et al., used a different cut-off of 2 or higher, which maximized sensitivity. Bennett et al., [28] found that PHQ-2 had a sensitivity of 93%, 82%, and 80%, and specificity of 75%, 80%, and 86%, at 15 and 30 weeks gestational age and 6–16 weeks postpartum, respectively. However, they found low PPVs (44, 24, and 30% respectively), similarly to our antenatal PPV. Furthermore, Bennett et al., adopted a different PHQ-2 response format (dichotomous yes/no, where responding yes to either of the items was considered a positive result), which maximised sensitivity and

specificity [29] and a sample with low education levels. It should be noted that it was found that PHQ-2 sensitivity was higher for women who went on beyond high school education than for those with a high school education or less [27]. Chae et al., [30], finally, found that the sensitivity of the PHQ-2 was 100% and the specificity was 79.3% among postpartum women attending a family multi-ethnic medicine residency centre. However, also in this case, the Authors adopted a dichotomous PHQ-2 response format which maximised sensitivity and specificity.

Overall, these studies [26–28, 30] suggest that lowering the cut-off point to 2 or higher will increase sensitivity. This, however, would come at the cost of lowered specificity given its inverse relationship with sensitivity. Moreover, at this lower cut-off the specificity is likely to be further reduced when the prevalence of depression is low [24, 29]. This suggests that the PHQ-2 at a cut-off point of 2 or more may have limited usefulness in the identification of women with depression in primary or secondary care services, because in such contexts the prevalence of depression is likely to be low. The extent to which lowering the cut-off point would be a valid option depends on the prevalence of depression and the cost and availability of subsequent strategies, potentially burdensome in busy maternity settings, to further assess those who score positively on the initial screening.

## Strengths and limitations

The strengths of the present study include the use of a large sample and several maternity clinical centres throughout Italy. The limitations include a low prevalence of antenatal depression and the use of the EPDS as a reference standard instead of a structured diagnostic interview. However, our prevalence is only slightly less than and comparable to those reported by other Italian studies [3, 4], and the aim in the present study was not to evaluate the diagnostic accuracy of the PHQ-2 but rather its screening accuracy compared to a longer more accredited screening instrument. In other words, the aim in the present study was to investigate if the PHQ-2 was accurate enough to be used as a substitute for a longer screening tool (as would occur in clinical practice using the ICHOM recommended procedure), not for a diagnostic instrument which formally assessed major depression.

## Conclusion

This study found limited evidence regarding the use of the PHQ-2 as part of a perinatal multistage case finding strategy at the recommended cut-off point of ≥ 3 [23, 37, 42]. Other Authors have shown that similar ultra-rapid screening instruments, such as the Whooley questions [20], had very low sensitivity when routinely administered in early pregnancy [46].

Maternity and primary care services require simple, quick screening tools to know who to refer. The present study calls into question the appropriateness of the PHQ-2, which aligns with the literature raising questions about the validity of the Whooley questions. The findings in this study may impact on service provision and making decisions about which screening instruments to use. It suggests that more comprehensive instruments may be needed as a first line. In our opinion, directly administering a longer screening instrument such as the EPDS may be the best option. Should this be difficult in a busy maternity setting, a tablet or paper could represent a useful alternative [46].

## Supporting information

**S1 Raw data. Data set on the accuracy of the Patient Health Questionnaire-2 in detecting depression among perinatal women in Italy.**
(XLSX)

## Acknowledgments

We would like to express our special thanks to PsyD Enrica Carluccio for her valuable technical assistance and contribution for the study.

## Author Contributions

**Conceptualization:** Antonella Gigantesco.

**Data curation:** Antonella Gigantesco.

**Formal analysis:** Fiorino Mirabella.

**Investigation:** Loredana Cena.

**Methodology:** Antonella Gigantesco.

**Project administration:** Loredana Cena.

**Supervision:** Alberto Stefana.

**Writing – original draft:** Antonella Gigantesco.

**Writing – review & editing:** Antonella Gigantesco, Gabriella Palumbo, Laura Camoni, Alice Trainini, Alberto Stefana.

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
