## [Editor Report · Decision Letter 0]

14 Jul 2021

PONE-D-21-08926

The screening accuracy of the Patient Health Questionnaire-2 in detecting depression among perinatal women in Italy

PLOS ONE

Dear Dr. Gigantesco,

Thank you for submitting your manuscript to PLOS ONE. After careful consideration, we feel that it has merit but does not fully meet PLOS ONE’s publication criteria as it currently stands. Therefore, we invite you to submit a revised version of the manuscript that addresses the points raised during the review process.

We look forward to receiving your revised manuscript.

Kind regards,

Angela Lupattelli, PhD

Academic Editor

PLOS ONE

Journal Requirements:

3. PLOS ONE does not conduct masked peer review. Please add references into manuscripts.

---

## [Author Response · Author response to Decision Letter 0]

27 Aug 2021

We address the reviewers/Editor concerns in the Response to Reviewers file, and corresponding changes have been made to the manuscript.

---

## [Decision Letter · Decision Letter 1]

13 Sep 2021

PONE-D-21-08926R1The screening accuracy of the Patient Health Questionnaire-2 in detecting depression among perinatal women in ItalyPLOS ONE

Dear Dr. Gigantesco,

Thank you for submitting your manuscript to PLOS ONE. After careful consideration, we feel that it has merit but does not fully meet PLOS ONE’s publication criteria as it currently stands. Therefore, we invite you to submit a revised version of the manuscript that addresses the points raised during the review process.

We look forward to receiving your revised manuscript.

Kind regards,

Angela Lupattelli, PhD

Academic Editor

PLOS ONE

Reviewers' comments:

Reviewer's Responses to Questions

**Comments to the Author**

1. If the authors have adequately addressed your comments raised in a previous round of review and you feel that this manuscript is now acceptable for publication, you may indicate that here to bypass the “Comments to the Author” section, enter your conflict of interest statement in the “Confidential to Editor” section, and submit your "Accept" recommendation.

Reviewer #1: (No Response)

Reviewer #2: All comments have been addressed

2. Is the manuscript technically sound, and do the data support the conclusions?

Reviewer #1: Partly

Reviewer #2: Yes

3. Has the statistical analysis been performed appropriately and rigorously? 

Reviewer #1: Yes

Reviewer #2: Yes

4. Have the authors made all data underlying the findings in their manuscript fully available?

Reviewer #1: Yes

Reviewer #2: Yes

5. Is the manuscript presented in an intelligible fashion and written in standard English?

Reviewer #1: Yes

Reviewer #2: No

6. Review Comments to the Author

Reviewer #1: Thank you for inviting me to review this manuscript. It reads well, and the analyses presented are accurate. However, there is a major limitation of this study; the assessment of criterion validity of PHQ-2 must be done against a gold standard i.e. diagnostic interviews by mental health specialists.

The EPDS and PHQ-2 could be compared to establish convergent validity but not criterion validity. This is because EPDS itself is a screening instrument and that too not with a perfect sensitivity and specificity.

The authors mention that psychodiagnostic interviews have been conducted by a clinical psychologist, as part of the study. Perhaps they could revise the analyses using the interview data rather than the EPDS one. I am sorry I could not be much encouraging of the choice of EPDS as a gold standard comparator and added further to the investigators' work. However, with updated analyses, the quality of the manuscript will be significantly improved.

I look forward to reading your revised work! Thank you for your excellent work.

Best wishes,

Reviewer #2: This study looks at the validity of the PHQ-2 as a screening tool for identifying perinatal depression. It uses the EDPS as a reference standard, rather than a more robust clinical interview. This approach facilitated the involvement of a large number of women across a wide geographical area. The study finds that the PHQ-2 may be inadequate in terms of sensitivity and overall accuracy in detecting maternal depression, as defined by a positive EPDS score. This study could contribute to the existing limited evidence regarding the diagnostic utility of the PHQ-2 in perinatal depression.

This study’s findings may inform policy decisions for screening for perinatal mental illness among the maternity population. Maternal depression often goes undetected and it is important to have a way of identifying at risk women in the general maternity population. There are many relatively new specialist perinatal mental health services being developed internationally, and referrals rely on perinatal illness being effectively picked up in Primary Care and maternity settings. This paper is a useful contribution to inform the referral process for these services.

The paper is difficult to read, with the author at times using confusing, ambiguous and superfluous phrases. This often obscures the meaning. Shorter, more precise sentences throughout would make it easier to read.

Overall, I think this is an interesting paper that may warrant publication with changes to the way it is written to convey its message and context more clearly.

Specific comments according to subheading:

Title (page 1)

1. Fine – although it could be more bold for impact, for instance the title could question the validity of using the PHQ-2 as a screening tool in perinatal depression

Abstract (page 2)

2. Last line confusing. The PHQ-2 performance suggested that it has insufficient sensitivity and discriminatory power, and may be inadequate as a screening tool for maternal depression.

Introduction (page 3)

3. Paragraph 1 - Remove ‘indeed, among other things’. Perinatal depression is associated with a range of adverse outcomes (neonatal outcomes, substance and alcohol abuse, poor attendance at antenatal care, bonding, family, long-term emotional and cognitive outcomes for the child).

4. Paragraph 2 - The clinical management of perinatal depression is evidence-based and includes pharmacological and psychological approaches. The treatments mentioned here in brackets are not internationally recognised - are they names of group/psychological therapies for women? Given that this paper has international interest I would make sure treatment interventions are described.

5. Paragraph 4 - Consider referencing the literature about the Whooley questions which questions the utility and accuracy of ultra-rapid screening tools (mentioned briefly in the conclusion). The NICE guidelines in 2007 were made in the absence of validation studies in the perinatal population. There is limited evidence for the use of the Whooley questions as a screening tool for maternal depression, only having been validated in small perinatal populations with a wide range of sensitivities found.

Methods (page 4)

6. More detail to allow replicability.

Results (page 5)

7. First paragraph - Sentence does not need to be in brackets, can be part of the text. Were all the questionnaires completed in full? What happened to partially completed questionnaires?

8. First paragraph - Could calculate the statistical significant of associations, using (?) non-parametric testing. Then statements could be made regarding positive associations eg – such as ‘amongst pregnant women, screening positive for depression on the EPDS was associated with secondary or university level education’. This is interesting because women from lower educational/socioeconomic groups are less likely to access perinatal mental health care, perhaps women from those groups less likely to disclose MH difficulties in screening questionnaires. Perhaps outside the scope of this paper.

9. Second and third paragraph – confusingly worded.

Discussion (page 6)

10. Develop explanation for the recommended scoring cut off for PHQ2. Gjerdingen et al used a cut off of 2 and yielded higher sensitivity. Bennet et al used dichotomous scoring system (closer to the Whooley Questions) and yielded higher sensitivity.

11. Original PHQ2 validation study – Perhaps some exploration of the difference in sensitivities, why did they get a sensitivity of 82%, study design? Sample size?

12. In last paragraph, use of the word ‘sensitively’ referring to questioning style not helpful in a discussion about statistical sensitivity.

13. Maternity and primary care services require simple, quick screening tools to know who to refer. This paper calls in to question the appropriateness of the PHQ2, which aligns with the literature raising questions about the validity of the Whooley questions. The findings in this paper may impact on service provision and making decisions about which screening tools to use. It suggests that more comprehensive tools may be needed as a first line with potential cost/implementation implications.

Conclusion (page 8)

14. Shorter, more clearly written conclusion required. Introducing new topics that could have been brought in earlier in the paper.

Tables (page 14)

15. Table 2 - ‘At least probable minor depression’ - this is a confusing name for the category. Consider ‘screened positive for depression’ or ‘EDPS positive for depression’. Something clearer to be used consistently throughout the paper.

16. Table 2 – Prevalence is usually expressed as a proportion

7. PLOS authors have the option to publish the peer review history of their article (what does this mean?). If published, this will include your full peer review and any attached files.

Reviewer #1: **Yes: **Dr. Ahmed Waqas

Reviewer #2: No

---

## [Author Response · Author response to Decision Letter 1]

28 Oct 2021

Manuscript: The screening accuracy of the Patient Health Questionnaire-2 in detecting depression among perinatal women in Italy

Response to reviewers (responses are shown highlighted)

We address the reviewers concerns in this letter, and corresponding changes have been made to the manuscript.

Reviewer #1

Thank you for inviting me to review this manuscript. It reads well, and the analyses presented are accurate. However, there is a major limitation of this study; the assessment of criterion validity of PHQ-2 must be done against a gold standard i.e. diagnostic interviews by mental health specialists.

The authors mention that psychodiagnostic interviews have been conducted by a clinical psychologist, as part of the study. Perhaps they could revise the analyses using the interview data rather than the EPDS one. I am sorry I could not be much encouraging of the choice of EPDS as a gold standard comparator and added further to the investigators' work. However, with updated analyses, the quality of the manuscript will be significantly improved.

We thank the reviewer for his/her valuable comments and we agree that the assessment regarding the criterion validity of PHQ-2 should be carried out against a gold standard, i.e., a diagnostic instrument. However, our aim was not to assess the diagnostic accuracy of the PHQ-2 but rather its screening accuracy in an attempt to corroborate the notion that it could be used instead of longer screening instruments, such as the EPDS. In other words, the aim of the current analysis was to reduce the burden of screening for depression which may derive from an inaccurate pre-screen step, and not to diagnose major depressive disorders, which should be performed by mental health specialists in a subsequent supplementary phase.

Moreover, our study extends some of the work of other validation studies of the PHQ-2 that used the EPDS as we did (Bennett et al., 2008; Chae et al., 2012; Cutler et al., 2007; Slavin et al., 2020) as a reference standard. Our study may help establish the reliability of those studies that did not find relatively high sensitivities and specificities of the PHQ-2 compared with the EPDS (Cutler et al., 2007; Slavin et al., 2020). 

In the future, we would be happy to assess the criterion validity of the PHQ-2 against a diagnostic instrument. In the present study, even if we wanted to, we wouldn’t be able to do it because the interviews were not diagnostic interviews. We apologise because we inadequately described the interview as diagnostic. We have now realised that the term psycho-diagnostic was inappropriate and we have removed it. The interviews were semi-structured to elicit some information regarding current and past maternal experience with severe psychiatric conditions and use of psychotropic drugs. In particular, psychiatric conditions included positive psychotic (i.e., delusions and/or hallucinations), non-suicidal self-harm tendencies, suicidal ideation or substance abuse symptoms. Women that reported current or past delusions and/or hallucinations, self-harm tendency or suicidal ideation or substance abuse symptoms were excluded from the study and invited to be further assessed by a psychiatric consultation service. This was done because the interviews were primarily finalized to ascertain the eligible criteria for the participation of women to a subsequent study on the effectiveness of a psychological intervention.

The EPDS and PHQ-2 could be compared to establish convergent validity but not criterion validity. This is because EPDS itself is a screening instrument and that too not with a perfect sensitivity and specificity.

We agree. Alternatively, the classic approach was actually to establish convergent validity. However, we were not interested in detecting a linear correlation (Pearson or Spearman correlation) between the continuous scores of the instruments, as this correlation would have likely been weak to moderate and poorly informative. On the contrary, we were more interested in the percentages of false-negatives and false-positives. Using dichotomous variables (positive vs. negative results) instead of continuous variables, as an alternative, we could have calculated the percentages of exact agreements between the two instruments using contingency tables and the chi-squared distribution. However, this analysis would have produced similar results to an accuracy analysis. 

Reviewer #2 

The paper is difficult to read, with the author at times using confusing, ambiguous and superfluous phrases. This often obscures the meaning. Shorter, more precise sentences throughout would make it easier to read.

We tried our best in the present version of the manuscript. In this regard, we would like to specify that the article had been sent to a paid language editor at the time of its first submission.

Specific comments according to subheading:

Title (page 1)

1. Fine – although it could be more bold for impact, for instance the title could question the validity of using the PHQ-2 as a screening tool in perinatal depression.

We have now slightly modified the title highlighting the limited validity of using PHQ-2 in perinatal depression.

Abstract (page 2)

2. Last line confusing. The PHQ-2 performance suggested that it has insufficient sensitivity and discriminatory power, and may be inadequate as a screening tool for maternal depression.

Thank you very much for the suggestion. We have now replaced the previous phrase with the phrase you suggested.

Introduction (page 3)

3. Paragraph 1 - Remove ‘indeed, among other things’. Perinatal depression is associated with a range of adverse outcomes (neonatal outcomes, substance and alcohol abuse, poor attendance at antenatal care, bonding, family, long-term emotional and cognitive outcomes for the child).

We have now removed it.

4. Paragraph 2 - The clinical management of perinatal depression is evidence-based and includes pharmacological and psychological approaches. The treatments mentioned here in brackets are not internationally recognised - are they names of group/psychological therapies for women? Given that this paper has international interest I would make sure treatment interventions are described.

We have now referred to internationally recognized psychological/psychosocial interventions and we have omitted the interventions that we previously reported.

5. Paragraph 4 - Consider referencing the literature about the Whooley questions which questions the utility and accuracy of ultra-rapid screening tools (mentioned briefly in the conclusion). The NICE guidelines in 2007 were made in the absence of validation studies in the perinatal population. There is limited evidence for the use of the Whooley questions as a screening tool for maternal depression, only having been validated in small perinatal populations with a wide range of sensitivities found.

We have added some additional information regarding the validity of the Whooley questions.

Methods (page 4)

6. More detail to allow replicability.

We have added more details in the Procedure and participants section.

Results (page 5)

7. First paragraph - Sentence does not need to be in brackets, can be part of the text. 

It is now part of the text.

Were all the questionnaires completed in full? What happened to partially completed questionnaires?

We have added that information to the Procedure and participants section.

8. First paragraph - Could calculate the statistical significant of associations, using (?) non-parametric testing. 

The chi-square test (or Fisher exact test) was used to test for differences between women with positive EPDS and women without for each socio-demographic characteristic. We have now added this information in the statistical analysis section.

Then statements could be made regarding positive associations e.g. – such as ‘amongst pregnant women, screening positive for depression on the EPDS was associated with secondary or university level education’. 

This has now been done.

This is interesting because women from lower educational/socioeconomic groups are less likely to access perinatal mental health care, perhaps women from those groups less likely to disclose MH difficulties in screening questionnaires. Perhaps outside the scope of this paper. 

Thank you for your discussion. We focused on this issue in another paper that we recently submitted to another journal. We agree that lower educational/socioeconomic groups are less likely to access perinatal mental health care. In fact, the sample in this study had a higher level of education and better financial situation compared to the general population of Italian women. For example, in the general population, about 22% of those aged 25-64 and 33% of those aged 30-34 years have obtained a University degree.

9. Second and third paragraph – confusingly worded.

We have rephrased those paragraphs.

Discussion (page 6)

10. Develop explanation for the recommended scoring cut off for PHQ2. Gjerdingen et al used a cut off of 2 and yielded higher sensitivity. Bennet et al used dichotomous scoring system (closer to the Whooley Questions) and yielded higher sensitivity.

We have expanded our discussion in the paragraphs in which we had discussed the findings of Gjerdingen and Bennet at the end of the Discussion section.

11. Original PHQ2 validation study – Perhaps some exploration of the difference in sensitivities, why did they get a sensitivity of 82%, study design? Sample size?

We are unable to explain why the original study registered a sensitivity of 82%. However, we suppose that in that validation study, the physicians reviewed the PHQ-2 and asked some additional questions in order to clarify responses on the questionnaires of patients who they felt might have “false negative” PHQ-2 results, as they did previously in the PRIME-MD study (Spitzer RL, Williams JBW, Kroenke K et al. Validation and utility of a self-report version of PRIME-MD. JAMA 1999; 282 (18): 1737-1744), in which the Authors used the longer PHQ-9. This may have inflated the sensitivity. 

In general, the vast majority of available studies on the accuracy of the PHQ-2 reported lower sensitivity than that in the original validation study. Accordingly, a recent systematic review has shown that in settings such as primary care or some inpatient and outpatient specialty care, the PHQ-2 was up to 62% sensitive for a cut-off score of 3 or greater in studies that used fully structured interviews as reference standards (Levis et al., 2020). 

12. In last paragraph, use of the word ‘sensitively’ referring to questioning style not helpful in a discussion about statistical sensitivity.

We have now omitted that word and the phrase.

13. Maternity and primary care services require simple, quick screening tools to know who to refer. This paper calls in to question the appropriateness of the PHQ2, which aligns with the literature raising questions about the validity of the Whooley questions. The findings in this paper may impact on service provision and making decisions about which screening tools to use. It suggests that more comprehensive tools may be needed as a first line with potential cost/implementation implications.

We agree with this valid point and have now referred to it in the text. 

Conclusion (page 8)

14. Shorter, more clearly written conclusion required. Introducing new topics that could have been brought in earlier in the paper.

This has now been done, also using your valuable comments (see point 13).

Tables (page 14)

15. Table 2 - ‘At least probable minor depression’ - this is a confusing name for the category. Consider ‘screened positive for depression’ or ‘EDPS positive for depression’. Something clearer to be used consistently throughout the paper.

Thank you. This has now been done.

16. Table 2 – Prevalence is usually expressed as a proportion

Thank you. This has now been done.

---

## [Editor Report · Decision Letter 2]

15 Nov 2021

The limited screening accuracy of the Patient Health Questionnaire-2 in detecting depression among perinatal women in Italy

PONE-D-21-08926R2

Dear Dr. Gigantesco,

We’re pleased to inform you that your manuscript has been judged scientifically suitable for publication and will be formally accepted for publication once it meets all outstanding technical requirements.

Kind regards,

Angela Lupattelli, PhD

Academic Editor

PLOS ONE

---

## [Editor Report · Acceptance letter]

17 Nov 2021

PONE-D-21-08926R2 

The limited screening accuracy of the Patient Health Questionnaire-2 in detecting depression among perinatal women in Italy 

Dear Dr. Gigantesco:

I'm pleased to inform you that your manuscript has been deemed suitable for publication in PLOS ONE. Congratulations! Your manuscript is now with our production department. 

Kind regards, 

on behalf of

Dr. Angela Lupattelli 

Academic Editor

PLOS ONE